# Impact of ENSO regimes on developing and decaying phase precipitation during rainy season in China

- 3 Qing Cao<sup>1</sup>, Zhenchun Hao<sup>1</sup>, Feifei Yuan<sup>1</sup>, Zhenkuan Su<sup>1</sup>, Ronny Berndtsson<sup>2</sup>, Jie Hao<sup>3</sup>, Tsring Nyima<sup>4</sup>
- 4 <sup>1</sup>State Key Laboratory of Hydrology Water Resources and Hydraulic Engineering, Hohai University, Nanjing, 210098, China
- 5 <sup>2</sup>Department of Water Resources Engineering and Center for Middle Eastern Studies, Lund University, Lund, PO Box 118, SE-
- 6 221 00, Sweden
- 7 <sup>3</sup>Nanjing Hydraulic Research Institute, Nanjing, 210098, China
- 8 <sup>4</sup>Investigation Bureau of hydrology and water resources in Ali of Tibet Autonomous Region, Tibet, 859000, China
- 9 Correspondence to: Zhenchun Hao (zhenchunhao@163.com) and Feifei Yuan (ffei.yuan@gmail.com)

10 Abstract. This study investigated the influence of five El Niño-Southern Oscillation (ENSO) types (i.e., Central Pacific Warming 11 (CPW), Eastern Pacific Cooling (EPC), Eastern Pacific Warming (EPW), conventional ENSO, and ENSO Modoki) on rainy-season 12 precipitation in China. The multi-scale moving t-test was applied to determine the onset and withdrawal of rainy season. Results 13 showed that there is a higher probability for flooding during decaying CPW and EPW phases in most parts of China with a largest 14 precipitation anomaly reaching 30% above average precipitation. Developing EPW could trigger droughts over large areas in China 15 with 10-30% lower than average precipitation in most areas. Conventional El Ni ño in the developing phase had the largest influence 16 on ENSO-related precipitation among developing ENSO and ENSO Modoki regimes. Decaying ENSO also showed larger effect 17 on the occurrence of drought and flood, compared to decaying ENSO Modoki. The difference between rainy-season precipitation 18 under various ENSO regimes may be attributed to the combined influence of anti-cyclone in the western North Pacific and the 19 Indian monsoon. Stronger monsoon and anti-cyclone are associated with enhanced rainy-season precipitation. The results suggest a 20 certain predictability of rainy-season precipitation related to ENSO regimes.

#### 21 1. Introduction

El Niño-Southern Oscillation (ENSO) is one of the most important factors affecting precipitation, which has been achieved urgent attention worldwide (Li et al., 2016;Wang et al., 2006;Preethi et al., 2015;Yuan et al., 2016a;Yuan et al., 2016b;Zaroug et al., 2014;Brigode et al., 2013). Many researchers have studied various aspects of ENSO-based precipitation, such as seasonal precipitation and extreme precipitation. Rainy season characteristics, however, are less considered, which are of immense significance to rain-fed agriculture in many countries like China. Reliable prediction of onset and withdrawal of rainy season will assist on-time preparation of farmlands and is significant to ecosystem (Omotosho et al., 2000;Marteau et al., 2011). In addition, rainy season is a period when it is easier for flooding and rainy-season precipitation could provide certain predictability for flood

occurrence. China is an ENSO-sensitive country and prone to flood and drought occurrence. Thus, it is significant to investigate 30 Chinese rainy-season precipitation under ENSO regimes. Cai (2003) observed similar inter-decadal oscillation and abrupt variations 31 between rainfall of rainy season in Fujian and Nino3 SST. Lu (2005) pointed out that rainfall in the rainy season in North China is 32 related to sea surface temperature anomalies (SSTA) in the equatorial eastern Pacific and negative (positive) SSTA could trigger 33 heavier (lighter) rainy-season precipitation. However, such studies mainly concentrated on regional scales and single ENSO mode, rather than continental scale and various ENSO regimes, which is important for overall understanding of relationship between ENSO 34 35 and Chinese rainy-season precipitation. In order to decipher this, it is necessary to explore the spatial pattern of precipitation during 36 the rainy season under various ENSO regimes at the continental scale in China. 37 Different types of ENSO regimes have been demonstrated based on the Pacific spatial pattern SSTA (Kao and Yu, 2009;Larkin

and Harrison, 2005; Ashok et al., 2007; Trenberth, 1997; Tedeschi et al., 2013; Kim et al., 2009). Conventional ENSO episodes, including El Niño (EN) and La Niña (LN), are defined based on SST anomalies in the NINO3.4 region, and El Niño is mainly 39 40 characterized by East Pacific warming in the cold tongue of the East Pacific ocean (Kim et al., 2009). Several researchers have 41 identified different episodes of SST in the Pacific, such as the central Pacific warming and east Pacific cooling (Larkin and Harrison, 42 2005; Weng et al., 2007; Kao and Yu, 2009). Kim et al. (2009) divided ENSO into three types, i.e., Central Pacific Warming (CPW), Eastern Pacific Cooling (EPC), and Eastern Pacific Warming (EPW). The division of ENSO is also based on SSTA in NINO 3, 43 44 NINO 3.4, and NINO 4 regions. Ashok et al. (2007) introduced a new type of ENSO event, ENSO Modoki, which is different from 45 conventional ENSO. ENSO Modoki is characterized by positive SSTA in the central Pacific, bounded by negative SSTA in the 46 western and eastern Pacific.

ENSO and ENSO Modoki have different influence on precipitation (Ashok et al., 2007;Ashok et al., 2009;Weng et al., 2007;Taschetto and England, 2009). Zhang et al. (2016a) pointed out that CPW, EPC, and EPW regimes showed various 48 49 performance on seasonal precipitation over the Huaihe River Basin. Precipitation below average usually occurs in southern China 50 in ENSO Modoki years, whereas the conventional ENSO tends to imply precipitation above average (Zhang et al., 2014b). In 51 contrast, enhanced precipitation over the Huaihe River Basin often occurs during decaying El Niño Modoki events in summer, 52 whilst reduced precipitation signals are found in the corresponding season in the decaying year of El Niño (Feng et al., 2011). It can 53 be seen that the influence of ENSO regimes on precipitation varies in different parts of China. The National Climate Center (NCC) 54 succeeded in predicting the severe flood over the Yangtze River basin in the typical El Ni ño year of 1997-1998. Nonetheless, NCC 55 failed to predict the enhanced precipitation in the Huaihe River basin in 2002-2003, since it was an El Ni ño Modoki year rather than a conventional El Niño. This highlights the significance of correct distinguishing between ENSO and ENSO Modoki. 56 57 Different performance of precipitation under various ENSO regimes is associated with atmospheric circulation and monsoon

(Tedeschi et al., 2013;Feng et al., 2010;Cai et al., 2010;Black et al., 2003;Chang et al., 2001;Zhang et al., 2014a;Onyutha and
Willems, 2015). Wu et al. (2003) explained the physical mechanism of links between precipitation and SSTs through features of

60 atmospheric circulation. Wang et al. (2004) pointed out that the local onset of rainy season in the South China Sea is related to mean 61 summer monsoon onset. Cai et al. (2010) argued that a rainfall reduction in southeast Queensland in Australia is related to an 62 eastward shift in the Walker circulation. Feng et al. (2011) pointed out that China rainfall anomalies were mainly due to anomalous anti-cyclonic flow in the western North Pacific associated with El Niño Modoki and El Niño events. Gerlitz et al. (2016) argued 63 64 that ENSO-induced precipitation variability in tropical regions is directly associated with the atmospheric circulation. The atmospheric circulation and monsoon have different influence on two types of ENSO (Feng and Li, 2013;Zhang et al., 2011;Zhou 65 and Chan, 2007). As a consequence, the investigation of atmospheric circulation and monsoon is used to explain different 66 67 performance of rainy-season precipitation anomalies under various ENSO regimes in this study.

The influence of ENSO and ENSO Modoki regimes on Chinese precipitation has been studied intensively. However, research 69 has been limited to the comparison of impacts of developing (decaying) ENSO and ENSO Modoki on precipitation at the regional 70 scale in China. Therefore, this study aims to improve our understanding of ENSO-induced precipitation during rainy season and to 71 explore the effect of five important ENSO types (i.e., CPW, EPC, EPW, ENSO and ENSO Modoki) in the developing and decaying 72 phase on the continental scale precipitation. The multi-scale moving t-test method was applied to determine the onset and withdrawal 73 of the rainy season. The underlying causes of the spatial patterns of rainy-season precipitation were analyzed by the variability of 74 atmospheric circulation in the western North Pacific (WNP) together with monsoon. Consequently, the paper is organized as follows. 75 Section 2 describes the study area and used data. Section 3 shows the methodology for determining the rainy season and the 76 definition of ENSO and ENSO Modoki. In Sect.4, we investigate and discuss the spatial distribution of rainy-season precipitation 77 under different ENSO regimes in the developing and decaying phase and their underlying causes. The final section summaries the 78 main findings.

#### 79 2. Study area and data

China, located in middle latitude in East Asia (18 N-54 °N, 73 °E- 135 °E), is the most populous country in the world (Fig. 1), with
a population of over 1.381 billion and an area of approximately 9.6 million km<sup>2</sup>. Climate of China is mainly dominated by monsoon
climate and mountain plateau climate, which lead to pronounced rainfall differences among different seasons and regions.

Daily precipitation data from 1960 to 2015 at 536 observation stations in China were selected for this study. The data were obtained from China Meteorological Data Sharing Service System, and the data quality has been regularly checked. The locations of the observation stations are shown in Fig.1. The stations are distributed unevenly, with fewer stations in the northwestern part of China. Hence, we applied Kriging interpolation to induce a resolution of  $0.2^{\circ} \times 0.2^{\circ}$ .

The dataset of National Oceanic and Atmospheric Administration (NOAA) extended reconstructed SST was used to identify different types of conventional ENSO. ENSO Modoki index (EMI) was obtained from the Japan Agency for Marine Science and Technology. In addition, the National Centres for Environmental Prediction (NCEP)/ National Centres for Atmospheric Research

- (NCAR) reanalysis data were used to investigate underlying causes of the spatial pattern of precipitation under different ENSO
- regimes (Kalnay et al., 1996).

Figure 1: The spatial distribution of precipitation stations used in this study.

3. Methodology

## 95 **3.1 Determination of rainy season**

The onset and withdrawal of rainy season was determined by the multi-scale moving t-test method. This method is characterized by the detection of mutation points between two subsamples with equal size *n*, where *n* is the length of the subsample, (*n*=30, 31, ..., 182/183; 182/183 corresponds to half the value of length of one year 365/366). Theoretically, the length of subsamples in this study ranged between 1 and 182/183. However, as the onset or withdrawal of the rainy season, it will not be considered if the length of the subsample is one day or just several days when the abruption point is prominent. As a result, the length of the subsample is limited between 30 and 182/183. The determination of the mutation point can be described as (Fraedrich et al., 1997)  $t(n, i) = (\bar{x}_{i2} - \bar{x}_{i1})n^{1/2}(s_{i2}^2 + s_{i1}^2)^{-1/2}$ , (1)

where  $\bar{x}_{i1}$  and  $\bar{x}_{i2}$  defined as,

$$\bar{x}_{i1} = \sum_{j=i-n}^{i-1} \frac{x_j}{n}; s_{i1}^2 = \sum_{j=i-n}^{i-1} (x_j - \bar{x}_{i1})^2 / (n-1),$$
(2)