# Peer review of "Impact of ENSO regimes on developing and decaying phase precipitation during rainy season in China"

_Hydrology and Earth System Sciences, 2017_

## Referee Comment (RC1) · Anonymous Referee #1 · 4 May 2017

General Comments:

ENSO has huge influence on precipitation in east Asia. This paper investigated the impact of different ENSO regimes on rainy-season precipitation in China at the developing and decaying phases, and it also explored the possible physical mechanism of precipitation change from the large-scale atmospheric circulation aspect. It will contribute important new knowledge to the study of the spatiotemporal rainy-season precipitation variability in China under different ENSO types. There are significant opportunities for improving the paper, which are presented as the following points:

Major comments

1. In study area and data part, how did the authors select the precipitation stations for analysisïij§How about the data quality? Any missing data? As far as I know, there might

have much more precipitation stations in China Meteorological Data Sharing Service System. What are the screening conditions for the selected stations?

2. In the methodology part, this paper defined the CPW, EPC and EPW regimes based on the definition proposed by Kim et al. (2009), and presented the years dominated by CPW, EPC and EPW regimes. However, the determination of conventional ENSO and ENSO Modoki in this paper is judged by the rainy reason rather than the whole year based on SD values. Could you explain why did do like this?

3. Line 194: the authors need to present a brief introduction why 850-mb vector is selected for the analysis of composites of circulation.

4. In the Results and Discussion part, the climate in China is largely affected by East Asian monsoon, which determines the spatiotemporal patterns of precipitation. How could you explain the connections between the monsoon effects and precipitation anomalies under different ENSO types?

Minor comments 1. Line 13-14: higher than normal average precipitation doesn't always mean flooding unless you conduct hydrological modeling. I would use precipitation anomaly only rather than "flood". I suggest "the precipitation anomaly can reach up to 30% above average precipitation during decaying CPW and EPW phase......"
2. Line 25: what does "rainy season characteristics represent?" 3. Line 29: Provide references for the statement "china is an ENSO-sensitive country" 4. Line 30: delete "Chinese" 5. Line 31: Nino3 → Niño 3. Global check over the manuscript. 6. Line 53: suggest revise "in different parts of china" to "among locations in china" 7. Line 74-78: Suggest delete the introduction of paper structure. It is more like the statement in a report. 8. Line 81: delete "Climate of" 9. Line 106: "Mann-Kendall test at 0.05 confidence level". Do you mean significance level? 10. Line 130: delete "It is seen that" 11. Line 226: try not to use vague words like "seems to..." 12. Line 240: "most parts of china": this statement should be more specific at which part. It is not good to use vague words like "most" in research paper. 13. Line 241-242: "the positive and negative anomaly

ranges from 0 to 30%...", Is this change significant at a certain confidence level?

---

## Referee Comment (RC3) · Anonymous Referee #2 · 1 Jun 2017

This paper attempts to investigate the impact of ENSO regimes (CPW, EPC, EPW, conventional ENSO, and ANSO Modoki) on precipitation in China through studying PARS (precipitation anomaly index during rainy season).

My main concerns of this work is:

First of all, why authors have only focused on precipitation anomaly in their approach? The description is not convincing. What about composite wind vectors at 850 mb? Is it a common approach to use these two measures to evaluate ENSO intensity and direction? I think this part of the paper needs more description and the current shape is not convincing. Also, what was the advantage of using NOAA extended reconstructed SST?

Below are some minor/moderate comments:

1) Figures 5, 6, and 7 are not blue shaded but gray shaded.

2) Line 29, "China is an ENSO-sensitive country and prone . . . .", how so? a reference will help. As it is in introduction, it should be more convincing.

3) Line 66, "As a consequence, the investigation of . . ." there is a missing piece that can connect using composite wind vectors at 850 mb to the investigation of atmospheric circulation.

4) Line 115, "The definition of ENSO Modoki and conventional ENSO was demonstrated." Not clear to me what authors wanted to highlight by this.

5) Line 186, authors stated that "spatial patterns of PARS under ENSO regimes may not only be determined by ENSO but also by the combination of various drivers" is it a result/finding of this study? Is there any other study that supports the idea?

---

## Author Comment (AC1) · 11 Jun 2017

The authors wish to thank the editors and reviewers for their effort in reviewing our manuscript. We appreciate the constructive criticisms, and we hope the changes listed have made the manuscript suitable for publication.

**Referee 1**:

General Comments:

ENSO has huge influence on precipitation in east Asia. This paper investigated the impact of different ENSO regimes on rainy-season precipitation in China at the developing and decaying phases, and it also explored the possible physical mechanism of precipitation change from the large-scale atmospheric circulation aspect. It will contribute important new knowledge to the study of the spatiotemporal rainy-season precipitation variability in China under different ENSO types. There are significant opportunities for improving the paper, which are presented as the following points:

**Major comments**

1) *Comments:*

In study area and data part, how did the authors select the precipitation stations for analysis? How about the data quality? Any missing data? As far as I know, there might have much more precipitation stations in China Meteorological Data Sharing Service System. What are the screening conditions for the selected stations?

**Response**

Thanks for the comment. It is true that more precipitation stations are shown in China Meteorological Data Sharing Service System as compared to stations selected in this paper. The screening conditions for the selected stations are:

a)  Daily precipitation data at each station is deleted if the missing data is more than 5% per year.

b)  Missing data is interpolated by the average value of effective daily precipitation 3 days before and after the missing data.

c)  The data period for selected station is no less than 40 years.

2) *Comments:*

In the methodology part, this paper defined the CPW, EPC and EPW regimes based on the definition proposed by Kim et al. (2009), and presented the years dominated by CPW, EPC and EPW regimes. However, the determination of conventional ENSO and ENSO Modoki in this paper is judged by the

rainy reason rather than the whole year based on SD values. Could you explain why did do like this?

**Response**

Thanks for the comment. The difference is determined by different definitions of CPW, EPC, EPW and ENSO, ENSO Modoki.

a)  The definition of CPW, EPC and EPW

Niño 4 warming exceeding 1 standard deviation (SD), while Niño 3 stays below this range, for CPW; Niño 3 or Niño 3.4 cooler than 1 SD, for EPC; Niño 3 warming greater than 1 SD, for EPW(Kim et al., 2009). Warming and cooling events are defined based on the detrended SST anomaly index for August to October (Kim et al., 2009).

It can be pointed out that the determination of CPW, EPC and EPW is based on SSTA and SD indices for August to October. Hence, the three ENSO types can only be determined by years rather than rainy season.

b)  The definition of ENSO and ENSO Modoki

The conventional EN (LN), abbreviated as CEN (CLN), was defined as SSTA above (below) 0.7 SD (-0.7 SD) in the area of 5°N − 5°S, 90°W − 140 ̊W. Similarly, warm (cold) episodes of ENSO Modoki, abbreviated as MEN (MLN), was defined as EMI above (below) 0.7 SD (-0.7 SD).

SD in definitions of ENSO and ENSO Modoki represents the standard deviation of the specific period you select. Therefore, we can judge the type of ENSO and ENSO Modoki of the rainy season.

3)  *Comments:*

Line 194: the authors need to present a brief introduction why 850-mb vector is selected for the analysis of composites of circulation.

**Response**

Thanks for the comment. We agree that the reason why choosing 850-mb vector winds to analyze circulation and monsoon is missing. A brief introduction is presented after the paragraph at ***Page 3, Line 67*** as:

"850hpa wind variability is associated with SSTA in the equatorial Pacific and precipitation anomalies in China (Zhang et al., 1999;Zhou and Chan, 2007;Wang et al., 2004;Zhang et al., 2016b). Fan et al. (2013) pointed out that 850 hPa vector winds are related to the moisture transportation from western tropical Pacific to the subtropical region, which determines the precipitation over the Yangtze-Huai River

Valley region. Huang et al. (2004) and Zhang et al. (2014a) presented the atmospheric circulation and monsoon variability by the composite distribution of wind anomalies at 850 hpa in different phases of El Niño and La Niña to explain precipitation variation in China. Feng et al. (2011) compared the difference of 850 hPa wind anomalies in decaying ENSO and ENSO Modoki phases to explain the physical mechanism of seasonal precipitation variation in China. Hence, 850 hpa vector winds reflecting atmospheric circulation and monsoon variability is used to explore the underlying causes of precipitation anomalies in this study. "

4) *Comments:*

In the Results and Discussion part, the climate in China is largely affected by East Asian monsoon, which determines the spatiotemporal patterns of precipitation. How could you explain the connections between the monsoon effects and precipitation anomalies under different ENSO types?

**Response**

Thanks for the comment. We use the spatial patterns of 850-mb vector winds to explain the connections between the monsoon effects and precipitation anomalies under different ENSO types. The reason why 850-mb vector winds are selected is presented in comment 3. We have cited more references to further show the teleconnection by revising the paragraph at *Page 10 Line 219* as:

"Generally, stronger western and southwestern winds are related to increasing precipitation. It is in agreement with the research of Zhang et al. (1996) and Wang et al. (2000), who pointed out that southeastern and southwestern winds could substantially enhance the moisture transportation to China. Wu et al. (2003) also found that East Asian monsoon is positively related to precipitation variations, which is consistent with our result."

The explanation of the teleconnection can also been shown at *Page 10 Lines 194-196* as:
 "There is a strengthening of westerly and southwesterly wind in the decaying year of CPW (Fig.5b), which brings more moisture to China, compared to developing CPW (Fig.5a). This may explain the enhanced precipitation in decaying CPW (Fig.2b)."

at *Page 10 Lines 205-207* as:
 "Compared to developing CEN, developing MEN experienced reduced precipitation in western China and generally enhanced precipitation in eastern parts under the combined influence of stronger monsoon and weakened anti-cyclone (Fig.6a-b)."

at *Page 11 Lines 211-213* as:
 "The difference of wind composites between decaying CLN and MLN indicates similar configuration, with stronger westerly wind and anti-cyclone causing enhanced precipitation for decaying MLN. "

**Minor comments**

**1)** *Comments:*

Lines 13-14: higher than normal average precipitation doesn't always mean flooding unless you conduct hydrological modeling. I would use precipitation anomaly only rather than "flood". I suggest "the precipitation anomaly can reach up to 30% above average precipitation during decaying CPW and EPW phase"

**Response**

Thanks for the comment. We will use precipitation anomaly only rather than "flood" or "drought".

P. 1, it is written: < Results showed that there is a higher probability for flooding during decaying CPW and EPW phases in most parts of China with a largest precipitation anomaly reaching 30% above average precipitation. >

revise the paragraph at *Page 1, Lines 12-14* as:

"Results showed that the precipitation anomaly can reach up to 30% above average precipitation during decaying CPW and EPW phases."

P. 1, it is written: < Developing EPW could trigger droughts over large areas in China with 10-30% lower than average precipitation in most areas. >

revise the paragraph at *Page 1, Lines 14-15* as:

"Developing EPW could cause decreasing precipitation over large areas in China with 10-30% lower than average precipitation in most areas."

P. 1, it is written: < Decaying ENSO also showed larger effect on the occurrence of drought and flood, compared to decaying ENSO Modoki.>

revise the paragraph at *Page 1, Lines 16-17* as:

"Decaying ENSO also showed larger effect on precipitation anomalies, compared to decaying ENSO Modoki."

P. 6, it is written: < In summary, the CPW decaying phase (EPC developing phase) deserves more attention than the developing (decaying) phase, since it has higher possibility to trigger flooding.>

revise the paragraph at *Page 6, Lines 143-144* as:

"In summary, the CPW decaying phase (EPC developing phase) deserves more attention than the developing (decaying) phase, since it show more prominent wet signals."

P. 8, it is written: < Flooding or drought is more easily triggered for the warm episodes of conventional ENSO, in comparison to the other three regimes.>

revise the paragraph at *Page 8, Lines 167* as:

"Wet or dry signals are more easily shown for the warm episodes of conventional ENSO, in comparison to the other three regimes."

P. 8, it is written: < Most parts of China presented increasing precipitation for decaying CEN, with more than 30% above average precipitation identified in north China, which is more likely to trigger flooding>

revise the paragraph at *Page 8, Lines 169-171* as:

"Most parts of China presented increasing precipitation for decaying CEN, with more than 30% above average precipitation identified in north China."

P. 13, it is written: < Conventional ENSO in the decaying phase is more likely to cause flooding and drought in comparison to the corresponding ENSO Modoki regimes.>

revise the paragraph at *Page 13, Lines 246-247* as:

"Conventional ENSO in the decaying phase is more likely to show wet and dry signals in comparison to the corresponding ENSO Modoki regimes."

2) *Comments:*

Line 25: what does "rainy season characteristics represent?"

P. 1, it is written: < Rainy season characteristics, however, are less considered, which are of immense significance to rain-fed agriculture in many countries like China. >

**Response**

Thanks for the comment. Rainy season characteristics represent onset, withdrawal and precipitation of the rainy season mentioned below. We have now revised the paragraph at *Page 1, Line 25* as:

"Rainy season characteristics (e.g., onset, withdrawal and precipitation of rainy season), however, are less considered, which are of immense significance to rain-fed agriculture in many countries like China."

**3) *Comments:***

Line 29: Provide references for the statement "china is an ENSO-sensitive country"

P. 2, it is written: < China is an ENSO-sensitive country and prone to flood and drought occurrence. >

**Response**

Thanks for the comment. We agree that the statement needs to be expanded to be made more clearly. We have now revised the paragraph at ***Page 2, Line 29*** as:

"China is an ENSO-sensitive country and prone to flood and drought occurrence (Zhang et al., 2016a;Feng et al., 2011;Feng et al., 2010;Wang and Wang, 2013;Zhang et al., 2014b;Feng and Li, 2011)"

**4) *Comments:***

Line 30: delete "Chinese"

P. 2, it is written: < Thus, it is significant to investigate Chinese rainy-season precipitation under ENSO regimes.>

**Response**

Thanks for reading thoroughly. We have now revised the paragraph at ***Page 2, Line 30*** as:

"Thus, it is significant to investigate rainy-season precipitation under ENSO regimes."

**5) *Comments:***

Line 31: Nino3 → Niño 3. Global check over the manuscript.

**Response**

Thanks for reading thoroughly. This point has been corrected.

**6) *Comments:***

Line 53: suggest revise "in different parts of China" to "among locations in China"

P. 2, it is written: < It can be seen that the influence of ENSO regimes on precipitation varies in different parts of China.>

**Response**

Again, thanks for reading thoroughly. We have now revised the paragraph at *Page 2, Line 53* as:

"It can be seen that the influence of ENSO regimes on precipitation varies among locations in china."

**7) Comments:**

Lines 74-78: Suggest delete the introduction of paper structure. It is more like the statement in a report.

**Response**

Thanks for the comment, we agree that it may be not appropriate to include the introduction of paper structure in a research article and we have corrected it.

**8) Comments:**

Line 81: delete "Climate of"

P. 3, it is written: <Climate of China is mainly dominated by monsoon climate and mountain plateau climate, which lead to pronounced rainfall differences among different seasons and regions.>

**Response**

Again, thanks for reading thoroughly. We have now revised the paragraph at *Page 3, Line 81* as:

"China is mainly dominated by monsoon climate and mountain plateau climate, which lead to pronounced rainfall differences among different seasons and regions."

**9) Comments:**

Line 106: "Mann-Kendall test at 0.05 confidence level". Do you mean significance level?

P. 5, it is written: < at 0.05 confidence level.>

**Response**

Thanks for the comment. We have now revised the paragraph at *Page 5, Line 106* as:

"at 0.05 significance level."

**10) Comments:**

Line 130: delete "It is seen that"

**Response**

Again, thanks for reading thoroughly. It has been corrected.

**11)** *Comments:*

Line 226: try not to use vague words like "seems to"

P. 11, it is written: <As a consequence, WNP anti-cyclone seems to have larger effect on East Asia precipitation on the inter-annual or inter-decadal scale,>

**Response**

Again, thanks for reading thoroughly. We have now revised the paragraph at *Page 11, Line 226* as:

"As a consequence, WNP anti-cyclone has larger effect on East Asia precipitation on the inter-annual or inter-decadal scale,"

**12)** *Comments:*

Line 240: "most parts of China": this statement should be more specific at which part. It is not good to use vague words like "most" in research paper.

P. 13, it is written: <It was found that most parts of China experience increasing precipitation for decaying CPW and EPW,>

**Response**

Thanks for the comment. We agree that it is not appropriate to use vague words in research paper, and "most parts of china" has been replaced by specific regions and locations in China. We have now revised the paragraph at *Page 13, Line 240* as:

"It was found that northwestern, central and southeastern China experience increasing precipitation for decaying CPW and EPW,"

**13)** *Comments:*

Lines 241-242: "the positive and negative anomaly ranges from 0 to 30%...", Is this change significant at a certain confidence level?

**Response**

Thanks for the comment. The significance level of precipitation anomaly of each precipitation station is truly considered. However, the paper has induced station data into grid data with a resolution of $0.2° \times$

0.2 ° by Kriging interpolation, because the stations are distributed unevenly. Hence, the spatial patterns of precipitation anomaly shown in this paper do not present the significance level. The significance level of stations are shown as Table. 1 (which is shown in the supplement PDF). The precipitation anomalies in most of selected stations are statistically significant at the 0.05 significance level, as shown in Table.1

**Table 1. Number of stations which are statistically significant or insignificant at the 0.05 significance level during various ENSO regimes.**

| Phase | The developing phase | | The decaying phases | |
|---|---|---|---|---|
| Number of stations | Statistically significant stations | Insignificant stations | Statistically significant stations | Insignificant stations |
| CPW | 480 | 56 | 484 | 52 |
| EPC | 475 | 61 | 454 | 82 |
| EPW | 429 | 107 | 487 | 49 |
| CEN | 435 | 101 | 475 | 61 |
| CLN | 439 | 97 | 409 | 127 |
| MEN | 464 | 72 | 476 | 60 |
| MLN | 451 | 85 | 459 | 77 |

*Reference*

Fan, L., Shin, S.-I., Liu, Q., and Liu, Z.: Relative importance of tropical SST anomalies in forcing East Asian summer monsoon circulation, Geophys. Res. Lett., 40, 2471-2477, doi:10.1002/grl.50494, 2013.

Feng, J., Wang, L., Chen, W., Fong, S. K., and Leong, K. C.: Different impacts of two types of Pacific Ocean warming on Southeast Asian rainfall during boreal winter, J. Geophys. Res. Atmos., 115, D24122, doi:10.1029/2010jd014761, 2010.

Feng, J., Chen, W., Tam, C. Y., and Zhou, W.: Different impacts of El Niño and El Niño Modoki on China rainfall in the decaying phases, Int. J. Climatol., 31, 2091-2101, doi:10.1002/joc.2217, 2011.

Feng, J., and Li, J.: Influence of El Niño Modoki on spring rainfall over south China, J. Geophys. Res., 116, D13102, doi:10.1029/2010jd015160, 2011.

Huang, R., Chen, W., Yang, B., and Zhang, R.: Recent advances in studies of the interaction between the East Asian winter and summer monsoons and ENSO cycle, Adv. Atmos. Sci., 21, 407-424, 2004.

Kim, H.-M., Webster, P. J., and Curry, J. A.: Impact of shifting patterns of Pacific Ocean warming on North Atlantic tropical cyclones, Science, 325, 77-80, doi:10.1126/science.1174062, 2009.

Wang, B., Wu, R., and Fu, X.: Pacific–East Asian teleconnection: how does ENSO affect East Asian climate?, J. Climate, 13, 1517-1536, 2000.

Wang, B., Zhang, Y., and Lu, M.: Definition of South China Sea monsoon onset and commencement of the East Asia summer monsoon, J. Climate, 17, 699-710, doi:10.1175/2932.1, 2004.

Wang, C., and Wang, X.: Classifying El Niño Modoki I and II by Different Impacts on Rainfall in Southern China and Typhoon Tracks, J. Climate, 26, 1322-1338, doi:10.1175/jcli-d-12-00107.1, 2013.

Wu, R., Hu, Z.-Z., and Kirtman, B. P.: Evolution of ENSO-related rainfall anomalies in East Asia, J. Climate, 16, 3742-3758, doi:10.1175/1520-0442(2003)016<3742:EOERAI>2.0.CO;2, 2003.

Zhang, Q., Wang, Y., Singh, V. P., Gu, X., Kong, D., and Xiao, M.: Impacts of ENSO and ENSO Modoki+A regimes on seasonal precipitation variations and possible underlying causes in the Huai River basin, China, J. Hydrol., 533, 308-319, doi:10.1016/j.jhydrol.2015.12.003, 2016a.

Zhang, R., Sumi, A., and Kimoto, M.: Impact of El Niño on the East Asian monsoon : A Diagnostic Study of the '86/87 and '91/92 Events, Journal of the Meteorological Society of Japan. Ser. II, 74, 49-62, doi:10.2151/jmsj1965.74.1_49, 1996.

Zhang, R., Sumi, A., and Kimoto, M.: A diagnostic study of the impact of El Niño on the precipitation in China, Adv. Atmos. Sci., 16, 229-241, 1999.

Zhang, R., Li, T., Wen, M., and Liu, L.: Role of intraseasonal oscillation in asymmetric impacts of El Niño and La Niña on the rainfall over southern China in boreal winter, Clim. Dyn., 45, 559-567, doi:10.1007/s00382-014-2207-4, 2014a.

Zhang, W., Jin, F. F., and Turner, A.: Increasing autumn drought over southern China associated with ENSO regime shift, Geophys. Res. Lett., 41, 4020-4026, doi:10.1002/2014GL060130, 2014b.

Zhang, W., Jin, F. F., Stuecker, M. F., Wittenberg, A. T., Timmermann, A., Ren, H. L., Kug, J. S., Cai, W., and Cane, M.: Unraveling El Niño's impact on the East Asian Monsoon and Yangtze River summer flooding, Geophys. Res. Lett., 43, 11375-11382, doi:10.1002/2016GL071190, 2016b.

Zhou, W., and Chan, J. C.: ENSO and the South China Sea summer monsoon onset, Int. J. Climatol., 27, 157-167, doi:10.1002/joc.1380, 2007.

---

## Author Comment (AC2) · 11 Jun 2017

The authors wish to thank the editors and reviewers for their effort in reviewing our manuscript. We appreciate the constructive criticisms, and we hope the changes listed have made the manuscript suitable for publication.

**Referee 2**:

**General Comments:**

This paper attempts to investigate the impact of ENSO regimes (CPW, EPC, EPW, conventional ENSO, and ENSO Modoki) on precipitation in China through studying PARS (precipitation anomaly index during rainy season).

**Major comments**

1) *Comments:*
First of all, why authors have only focused on precipitation anomaly in their approach?
The description is not convincing. What about composite wind vectors at 850 mb? Is it a common approach to use these two measures to evaluate ENSO intensity and direction? I think this part of the paper needs more description and the current shape is not convincing. Also, what was the advantage of using NOAA extended reconstructed

Thanks for the comment.
a) It is true that the reason why we have only focused on precipitation anomaly in our approach is unclear. A brief introduction is presented after the paragraph at *Page 5, Line 122* as:
"Precipitation anomaly can present the difference of precipitation between ENSO years and normal years and demonstrate the influence of ENSO regimes on precipitation more directly. Zhang et al. (2013) used precipitation anomaly index to explore the effect of ENSO on precipitation in the East River Basin, South China. Zhang et al. (2016a) investigated the influence of ENSO and ENSO Modoki on seasonal precipitation over the Huaihe River Basin by using precipitation anomaly index. "

b) We agree that the reason why choosing 850-mb vector winds to analyze circulation and monsoon is missing. A brief introduction is presented after the paragraph at *Page 3, Line 67* as:
"850hpa wind variability is associated with SSTA in the equatorial Pacific and precipitation anomalies in China (Zhang et al., 1999;Zhou and Chan, 2007;Wang et al., 2004;Zhang et al., 2016b). Fan et al. (2013) pointed out that 850 hPa vector winds are related to the moisture transportation from western tropical Pacific to the subtropical region, which determines the precipitation over the Yangtze-Huai River Valley region. Huang et al. (2004) and Zhang et al. (2014a) presented the atmospheric circulation and monsoon variability by the composite distribution of wind anomalies at 850 hpa in different phases of El Niño and La Niña to explain precipitation variation in China. Feng et al. (2011) compared the difference of 850 hPa wind anomalies in decaying ENSO and ENSO Modoki phases to explain the physical mechanism of seasonal precipitation variation in China. Hence, 850 hpa vector winds reflecting atmospheric circulation and monsoon variability is used to explore the underlying causes of precipitation anomalies in this study. "

Therefore, it is a common approach to use these two measures to evaluate ENSO intensity and direction.

c) The advantage of using NOAA extended reconstructed data

The Extended Reconstructed Sea Surface Temperature (ERSST) dataset is a global monthly sea surface temperature dataset derived from the International Comprehensive Ocean–Atmosphere Dataset (ICOADS). It is produced on a $2°\times2°$ grid with spatial completeness enhanced using statistical methods. The newest version of ERSST, version 4, used in this study, is based on optimally tuned parameters using the latest datasets and improved analysis methods (Liu et al., 2015;Huang et al., 2016). ERSST is suitable for long-term global and basin-wide studies, and smoothed local and short-term variations are used in the dataset.

**Minor comments**

**1)** *Comments:*

Figures 5, 6, and 7 are not blue shaded but gray shaded.

**Response**

Thanks for reading thoroughly. It has been corrected.

**2)** *Comments:*

Line 29, "China is an ENSO-sensitive country and prone : : :.", how so? a reference will help. As it is in introduction, it should be more convincing.

**Response**

Thanks for the comment. We agree that the statement needs to be expanded to be made more clearly. We have now revised the paragraph at *Page 2, Line 29* as:

"China is an ENSO-sensitive country and prone to flood and drought occurrence (Zhang et al., 2016a;Feng et al., 2011;Feng et al., 2010;Wang and Wang, 2013;Zhang et al., 2014b;Feng and Li, 2011)"

**3)** *Comments:*

Line 66, "As a consequence, the investigation of : : :" there is a missing piece that can connect using composite wind vectors at 850 mb to the investigation of atmospheric circulation.

**Response**

Thanks for the comment. We have added a piece after the paragraph at *Page 3, Line 67* as to connect using composite wind vectors at 850 mb to the investigation of atmospheric circulation.

"850hpa wind variability is associated with SSTA in the equatorial Pacific and precipitation anomalies in China (Zhang et al., 1999;Zhou and Chan, 2007;Wang et al., 2004;Zhang et al., 2016b). Fan et al. (2013) pointed out that 850 hPa vector winds are related to the moisture transportation from western tropical Pacific to the subtropical region, which determines the precipitation over the Yangtze-Huai River Valley region. Huang et al. (2004) and Zhang et al. (2014a) presented the atmospheric circulation and monsoon variability by the composite distribution of wind anomalies at 850 hpa in different phases of El Niño and La Niña to explain precipitation variation in China. Feng et al. (2011) compared the difference

of 850 hPa wind anomalies in decaying ENSO and ENSO Modoki phases to explain the physical mechanism of seasonal precipitation variation in China. Hence, 850 hpa vector winds reflecting atmospheric circulation and monsoon variability is used to explore the underlying causes of precipitation anomalies in this study. ”

**4) _Comments:_**

Line 115, "The definition of ENSO Modoki and conventional ENSO was demonstrated." Not clear to me what authors wanted to highlight by this.

**Response**

Thanks for the comment. The purposes to demonstrate the definition of ENSO and ENSO Modoki is:

a) Show the difference of ENSO and ENSO Modoki.

b) Facilitate readers and other researchers to have a better knowledge of the research process, and the judgment of ENSO and normal years is based on their definitions.

**5) _Comments:_**

Line 186, authors stated that "spatial patterns of PARS under ENSO regimes may not only be determined by ENSO but also by the combination of various drivers" is it a result/finding of this study? Is there any other study that supports the idea?

**Response**

Thanks for the comment. It is not a result of this study, there are other studies to support the idea, which has been presented at **_Page 8,_** Lines **_181-185_**.

"Xu et al. (2016) revealed that increasing autumn precipitation in southern China is due to the combined ENSO and Indian Ocean Dipole (IOD) events. Other researchers also concluded that IOD and ENSO have mutual impact on precipitation anomalies in China (Weng et al., 2011; Liu et al., 2009; Wu et al., 2012). Moreover, Pacific Decadal Oscillation, subtropical high, also influence the distribution of Chinese precipitation (Chan, 2005; Wang et al., 2008; Chang et al., 2000; Niu and Li, 2008; Ouyang et al., 2014)."

Reference:

Fan, L., Shin, S.-I., Liu, Q., and Liu, Z.: Relative importance of tropical SST anomalies in forcing East Asian summer monsoon circulation, Geophys. Res. Lett., 40, 2471-2477, doi:10.1002/grl.50494, 2013.

Feng, J., Wang, L., Chen, W., Fong, S. K., and Leong, K. C.: Different impacts of two types of Pacific Ocean warming on Southeast Asian rainfall during boreal winter, J. Geophys. Res. Atmos., 115, D24122, doi:10.1029/2010jd014761, 2010.

Feng, J., Chen, W., Tam, C. Y., and Zhou, W.: Different impacts of El Niño and El Niño Modoki on China rainfall in the decaying phases, Int. J. Climatol., 31, 2091-2101, doi:10.1002/joc.2217, 2011.

Feng, J., and Li, J.: Influence of El Niño Modoki on spring rainfall over south China, J. Geophys. Res., 116, D13102, doi:10.1029/2010jd015160, 2011.

Huang, B., Thorne, P. W., Smith, T. M., Liu, W., Lawrimore, J., Banzon, V. F., Zhang, H.-M., Peterson, T. C.,

and Menne, M.: Further exploring and quantifying uncertainties for extended reconstructed sea surface temperature (ERSST) version 4 (v4), J. Climate, 29, 3119-3142, doi:10.1175/JCLI-D-15-0430.1, 2016.

Huang, R., Chen, W., Yang, B., and Zhang, R.: Recent advances in studies of the interaction between the East Asian winter and summer monsoons and ENSO cycle, Adv. Atmos. Sci., 21, 407-424, 2004.

Liu, W., Huang, B., Thorne, P. W., Banzon, V. F., Zhang, H.-M., Freeman, E., Lawrimore, J., Peterson, T. C., Smith, T. M., and Woodruff, S. D.: Extended reconstructed sea surface temperature version 4 (ERSST. v4): Part II. Parametric and structural uncertainty estimations, J. Climate, 28, 931-951, doi:10.1175/JCLI-D-14-00007.1, 2015.

Wang, B., Zhang, Y., and Lu, M.: Definition of South China Sea monsoon onset and commencement of the East Asia summer monsoon, J. Climate, 17, 699-710, doi:10.1175/2932.1, 2004.

Wang, C., and Wang, X.: Classifying El Niño Modoki I and II by Different Impacts on Rainfall in Southern China and Typhoon Tracks, J. Climate, 26, 1322-1338, doi:10.1175/jcli-d-12-00107.1, 2013.

Zhang, Q., Li, J., Singh, V. P., Xu, C. Y., and Deng, J.: Influence of ENSO on precipitation in the East River basin, South China, Journal of Geophysical Research: Atmospheres, 118, 2207-2219, 2013.

Zhang, Q., Wang, Y., Singh, V. P., Gu, X., Kong, D., and Xiao, M.: Impacts of ENSO and ENSO Modoki+A regimes on seasonal precipitation variations and possible underlying causes in the Huai River basin, China, J. Hydrol., 533, 308-319, doi:10.1016/j.jhydrol.2015.12.003, 2016a.

Zhang, R., Sumi, A., and Kimoto, M.: A diagnostic study of the impact of El Nino on the precipitation in China, Adv. Atmos. Sci., 16, 229-241, 1999.

Zhang, R., Li, T., Wen, M., and Liu, L.: Role of intraseasonal oscillation in asymmetric impacts of El Niño and La Niña on the rainfall over southern China in boreal winter, Clim. Dyn., 45, 559-567, doi:10.1007/s00382-014-2207-4, 2014a.

Zhang, W., Jin, F. F., and Turner, A.: Increasing autumn drought over southern China associated with ENSO regime shift, Geophys. Res. Lett., 41, 4020-4026, doi:10.1002/2014GL060130, 2014b.

Zhang, W., Jin, F. F., Stuecker, M. F., Wittenberg, A. T., Timmermann, A., Ren, H. L., Kug, J. S., Cai, W., and Cane, M.: Unraveling El Niño's impact on the East Asian Monsoon and Yangtze River summer flooding, Geophys. Res. Lett., 43, 11375-11382, doi:10.1002/2016GL071190, 2016b.

Zhou, W., and Chan, J. C.: ENSO and the South China Sea summer monsoon onset, Int. J. Climatol., 27, 157-167, doi:10.1002/joc.1380, 2007.

---

## Author Comment (AC3) · 12 Jul 2017

**Impact of ENSO regimes on developing and decaying phase precipitation during rainy season in China**

Qing Cao[1], Zhenchun Hao[1], Feifei Yuan[1], Zhenkuan Su[1], Ronny Berndtsson[2], Jie Hao[3], Tsring Nyima[4]

[revised manuscript text omitted]

$$\bar{x}_{i2} = \sum_{j=i}^{i+n-1} \frac{x_j}{n}; s_{i2}^2 = \sum_{j=i}^{i+n-1}(x_j - \bar{x}_{i2})^2 /(n-1), \tag{3}$$

and $x_i$ is daily precipitation for Julian day $i$ within one year and for one station. $\bar{x}_{i1}$ and $\bar{x}_{i2}$ are the mean values of the subsamples before and after the Julian day $i$, respectively.

The t-value calculated above was normalized by the 0.01 test value showing in Eq. (4), which is equal to the result of Mann-Kendall test at 0.05 significanceconfidence level.

$$t_r(n,i) = t(n,i)/t_{0.01}(n),  \tag{4}$$

where $t_r(n,i)$ can be taken as the threshold to detect mutations. $t_r(n,i) > 1.0$ represents an increasing trend while $t_r(n,i) < 1.0$

is a decreasing trend. The onset of rainy season in this study was defined as the mutation point corresponding to a maximum $t_r(n,i)$

value. For this case, precipitation changes from a smaller to a higher value. Likewise, the withdrawal is defined as the changing point corresponding to a minimum $t_r(n,i)$ value.

**3.2 Classification of ENSO and ENSO Modoki regimes**

Three types of ENSO were classified based on the definition proposed by Kim et al. (2009). The years dominated by CPW, EPC, and EPW are listed in Table 1.

**Table 1. Years dominated by CPW, EPC, and EPW regimes during 1960-2015.**

| EPW | EPC | CPW |
|---|---|---|
| 1965,1972,1976,1982,1987,1997,2015 | 1964,1970,1973,1975,1988,1998,1999, 2007,2010,2011 | 1963,1969,1991,1994,2002,2004,2009 |

The definition of ENSO Modoki and conventional ENSO was demonstrated. Specifically, warm (cold) episodes of ENSO

Modoki, abbreviated as MEN (MLN), was defined as EMI above (below) 0.7 SD (-0.7 SD), where SD is the standard deviation (Ashok et al., 2007). $EMI = [SSTA]A - 0.5 \times [SSTA]B - 0.5 \times [SSTA]C$, where [SSTA]A, [SSTA]B, [SSTA]C represents the

SSTA in region $A(10°S - 10°N, 165°E - 140°W)$, region $B(15°S - 5°N, 110°W - 70°W)$ and region $C(10°S - 20°N, 125°E -$

$145°E)$,respectively. Likewise, the conventional EN (LN), abbreviated as CEN (CLN), was defined as SSTA above (below) 0.7

SD (-0.7 SD) in the area of $5°N - 5°S, 90°W - 140°W$ (Tedeschi et al., 2013). This definition gives an opportunity to judge the

ENSO type of the rainy season rather than the whole year, which is greater than definition proposed by Trenberth (1997).

**3.3 Precipitation anomaly index during rainy season (PARS)**

——Precipitation anomaly can present the difference of precipitation between ENSO years and normal years and demonstrate the influence of ENSO regimes on precipitation more directly. Zhang et al. (2013a) used precipitation anomaly index to explore the effect of ENSO on precipitation in the East River Basin, South China. Zhang et al. (2016a) investigated the influence of ENSO and

ENSO Modoki on seasonal precipitation over the Huaihe River Basin by using precipitation anomaly index.

Precipitation anomaly index is  defined as:

$$PARS_{ij} = \left( \frac{\overline{PRS_{ij}}}{\overline{PRSN_{ij}}} - 1 \right) \times 100\%, \tag{5}$$

where $PARS_{ij}$ denotes precipitation anomaly during rainy season at $i_{th}$ station in $j_{th}$ year; $\overline{PRS_{ij}}$ denotes mean daily precipitation during   rainy season at $i_{th}$ station in $j_{th}$ year, and $\overline{PRSN_{ij}}$ denotes mean daily precipitation during rainy season at $i_{th}$ station in $j_{th}$ normal year. The normal year refers to a year without ENSO event occurring.

**4.    Results and Discussion**

**4.1    Precipitation anomaly during rainy season (PARS) influenced by CPW, EPC, and EPW regimes**

The spatial variability of PARS under CPW, EPC and EPW regimes in the phase of developing and decaying years is presented in Fig.2. The distribution of precipitation anomaly is irregular over the whole area in the developing phase of CPW. The coastal regions in southeastern China that had the largest amount of rainy-season precipitation presented the largest decreasing trend, with the precipitation anomaly reaching 30% below average precipitation. The upper and middle reaches of the Yangtze River and the Yellow River showed decreasing precipitation, whereas the lower reaches had the opposite trend. The decaying CPW regime had relatively regular spatial pattern. More specifically, most parts of China presented increasing precipitation during rainy season, with the largest PARS being 20% above average precipitation. The distribution of PARS influenced by the decaying CPW is similar to that by the developing EPC, with shrinking extent of enhanced precipitation in central China for developing EPC. The distribution of PARS is similar as well in the two phases of EPC (Fig.2, second row), with precipitation above average in northwestern China and precipitation below average in northeastern China. The difference between the two phases lies in the increasing (decreasing) precipitation in southeastern China in the developing (decaying) phase. Nonetheless, developing and decaying EPW (Fig.2, third row) showed opposite spatial precipitation pattern. Most parts of China presented dry signals in the phase of developing EPW, which became stronger northwards, and more than 30% below average precipitation can be identified in north China. However, there is above average precipitation in most regions of China in the case of decaying EPW, with PARS values ranging between 0 and 30%. In summary, the CPW decaying phase (EPC developing phase) deserves more attention than the developing (decaying) phase, since it show more prominent wet signals.  Both phases are significant for the EPW regimes, due to the obvious dry (wet) signals shown in the developing (decaying) phase.

[Figure]

**Figure 2:** Spatial pattern for rainy-season precipitation anomaly (PARS) during the CPW (first row), EPC (second row), and EPW (third row) episodes in the phase of ENSO developing year (0) and decaying year (1). The sign "0" in the parentheses denotes ENSO developing year and "1" denotes decaying year.

**4.2   Precipitation anomaly during rainy season (PARS) impacted by ENSO and ENSO Modoki regime**

Figure 3 and 4 present precipitation anomalies during rainy season (PARS) for warm and cold episodes of conventional ENSO and ENSO Modoki in a developing phase, and a decaying phase, respectively. Precipitation increased in a band stretching from northwestern China to the coastal region in the southeast, with the largest precipitation anomaly (40%) occurring in southeast China under a developing CEN regime (Fig.3a). The dry condition is more severe northwards in central China, with PARS equal to about -30% in the northern parts of central China. Zhang et al. (2016b) concluded that strong El Niño events are associated with summer monsoon flooding over the Yangtze River, which is consistent with our results. The distribution of rainy-season precipitation for developing El Niño is also in agreement with the research by Zhang et al. (2011). Northern China had opposite PARS pattern for developing El Niño Modoki, in comparison to developing CEN (Fig.3b). Nonetheless, the two phases showed similar precipitation distribution, with reduced precipitation in central China (approximately -10%) and enhanced precipitation in southern China. Typically, developing CEN demonstrated more obvious wet or dry signals compared to MEN. Moreover, the wet and dry condition for developing CEN is the most serious among all ENSO and ENSO Modoki regimes in both developing and decaying phases, with the largest precipitation anomaly reaching 50% above average precipitation amount and lowest 30% below. This means that developing CEN should be paid urgent attention for flooding and drought monitoring. The spatial pattern of PARS for developing CLN presented similar signals with developing MEN, with a shorter wet precipitation band in northern China for developing CLN (Fig.3c). The increased precipitation was shifted westwards for the developing MLN, compared to cold episodes of conventional ENSO (Fig.3d). ENSO and ENSO Modoki regimes in the developing phase presented various distribution of precipitation anomalies. Wet or dry signals are more easily shown for the warm episodes of conventional ENSO, in comparison to the other three regimes.  Similar patterns of PARS for developing CLN and MEN is suggested to be further studied.

Decaying ENSO and ENSO Modoki years showed different features of PARS (Fig.4). Most parts of China presented increasing precipitation for decaying CEN, with more than 30% above average precipitation identified in north China  (Fig.4a). The decaying phase of MEN (Fig.4b) presented shrinking extent of enhanced precipitation, which was condensed in the central parts of China, ranging between 0 and 10%. The result is consistent with conclusions from Feng et al. (2011), who found obvious rainfall anomalies in southern China for decaying El Niño and no prominent rainfall variations in the corresponding phase of El Niño Modoki. In terms of the cold episodes of ENSO (Fig.4c), approximately 95% of China showed dry signals, and the condition was more serious eastwards, being 30% below average precipitation amount. We can see that the spatial pattern of PARS for the decaying CLN is opposite to that of CEN. Decaying MLN (Fig.4d) showed larger extent of enhanced precipitation in a band stretching from western China to parts of the Yellow River Basin, in comparison to CLN. In conclusion, the decaying phases of conventional ENSO showed more obvious wet or dry signals compared to ENSO Modoki, with most parts of China displaying increasing (decreasing) precipitation for the CEN (CLN).

This study analyzed spatial patterns of precipitation under different ENSO regimes, since ENSO is the leading driver of precipitation anomaly in China (Xiao et al., 2015). Xu et al. (2016) revealed that increasing autumn precipitation in southern China is due to the combined ENSO and Indian Ocean Dipole (IOD) events. Other researchers also concluded that IOD and ENSO have mutual impact on precipitation anomalies in China (Weng et al., 2011;Liu et al., 2009;Wu et al., 2012). Moreover, Pacific Decadal Oscillation, subtropical high, also influence the distribution of Chinese precipitation (Chan, 2005;Wang et al., 2008;Chang et al.,

2000;Niu and Li, 2008;Ouyang et al., 2014). As a result, the spatial patterns of PARS under ENSO regimes may not only be determined by ENSO, but also by the combination of various drivers, which ought to be studied further.

[Figure]

**Figure 3: Spatial pattern of precipitation anomalies during rainy season (PARS) during developing (0) conventional ENSO and ENSO**
**Modoki events.**

[Figure]

**Figure 4: Spatial pattern of precipitation anomalies during rainy season (PARS) for decaying (1) conventional ENSO and ENSO Modoki events.**

**4.3    Composites of circulation**

Figure 5 presents the composites of 850-mb vector wind for three types of ENSO. There is a strengthening of westerly and southwesterly wind in the decaying year of CPW (Fig.5b), which brings more moisture to China, compared to developing CPW (Fig.5a). This may explain the enhanced precipitation in decaying CPW (Fig.2b). The difference between developing and decaying EPC (Fig.5c-d) lies in the shift of anti-cyclonic flow in the western part of North Pacific (WNP). The eastward anti-cyclone for the decaying EPC weakened the transportation of moisture in eastern China and caused reduced precipitation (Fig.2d). The decaying EPW (Fig.5f) experienced stronger western and southwestern wind but weakened anti-cyclone compared to the developing phase (Fig.5e). The WNP anti-cyclone could bring plentiful moisture to China, so weakened anti-cyclonic flow will cause reduced precipitation (Feng et al., 2011). However, most parts of China presented wetter signals in the phase of decaying EPW in comparison to developing EPW. Therefore, it can be pointed out that the India monsoon plays a more significant role in the formation of rainy-season precipitation during EPW phases compared to the atmospheric circulation.

Figure 6 shows the underlying causes of different performance of conventional ENSO and ENSO Modoki in the developing phase by analysis of the 850-hp wind. Compared to developing CEN, developing MEN experienced reduced precipitation in western China and generally enhanced precipitation in eastern parts under the combined influence of stronger monsoon and weakened anti-cyclone (Fig.6a-b). Stronger anti-cyclonic flow in the phase of developing La Niña Modoki (Fig.6d) may cause the enhanced precipitation in western parts of China compared to conventional La Niña regime in developing years (Fig.6c).

The wind composites of warm and cold episodes of decaying ENSO and ENSO Modoki are presented in Fig.7. Compared to decaying CEN, the wet signal of precipitation is weaker in the decaying year of MEN, which may be attributed to the weakened anti-cyclonic flow in WNP and western winds for the decaying MEN. The difference of wind composites between decaying CLN and MLN indicates similar configuration, with stronger westerly wind and anti-cyclone causing enhanced precipitation for decaying MLN.

In summary, westerly winds seem to play more significant role in the phase of CPW and EPW, while developing La Niña and La Niña Modoki are dominated by the anti-cyclone. The spatial pattern of PARS is the reflection of combined influence of westerly winds and anti-cyclonic flow for the EPC and decaying ENSO and ENSO Modoki regimes.

It can be seen that the spatial pattern of precipitation during the rainy season in China is dominated by westerly winds from India and anti-cyclone in WNP, which is equivalent to the results by Dai and Wigley (2000), Feng and Li (2011), Wu et al. (2003). Generally, stronger western and southwestern winds are related to increasing precipitation. It is in agreement with the research of Zhang et al. (1996) and Wang et al. (2000), who pointed out that southeastern and southwestern winds could substantially enhance the moisture transportation to China. Wu et al. (2003) also found that East Asian monsoon is positively related to precipitation variations, which is consistent with our result. Likewise, the westward and stronger anti-cyclone is related to enhanced PARS. Wu et al. (2003) reported that the anomalous low-level anti-cyclone is determined by large-scale equatorial heating anomalies and local air-sea interactions. Westerlies and anti-cyclone are of dominant importance for the ENSO-induced precipitation during the rainy season. However, cyclonic flow may have larger influence on Chinese precipitation under certain circumstances. For example, the autumn drought in southwest China in 2009 was determined by a strong cyclone in WNP for ENSO Modoki (Zhang et al., 2013b).

Feng et al. (2011) also revealed that the WNP circulation is cyclonic in winter and then becomes weak in the following spring and anti-cyclonic flow in summer for El Niño Modoki. As a consequence, WNP anti-cyclone has larger effect on East Asia precipitation on the inter-annual or inter-decadal scale, but anti-cyclone and cyclone are both crucial for the determination of precipitation on the annual or smaller scale.

[Figure]

**Figure 5:** Composites of 850-mb vector wind for mainland China during CPW, EPC and EPW developing (0) and decaying (1) phases. Arrows show the direction of wind (m/s); grey shaded areas denote wind speed above 3 m/s.

[Figure]

**Figure 6:** Composites of 850-mb vector wind for mainland China during ENSO and ENSO Modoki developing (0) phases. Arrows show the direction of wind (m/s); grey shaded areas denote wind speed above 3 m/s.

[Figure]

**Figure 7:** Composites of 850-mb vector wind for mainland China during ENSO and ENSO Modoki decaying (1) phases. Arrows show the direction of wind (m/s); blue shaded areas denote wind speed above 3 m/s.

**5. Conclusion**

This study investigated the distribution of PARS under various ENSO types in developing and decaying phases and their underlying causes. It was found that northwestern, central and southeastern China experience increasing precipitation for decaying CPW and EPW, and positive precipitation anomaly ranges from 0 to 30% due to the stronger westerly and southwesterly winds. The developing phase of EPW presents overall negative rainy-season precipitation anomalies in China with more than 30% below average precipitation identified in many parts of the country, which is a result from weak westerly winds. The different spatial distribution of rainy-season precipitation under developing and decaying ENSO and ENSO Modoki regimes was also examined. Conventional El Niño in developing years showed larger influence on precipitation during rainy season in China as compared to developing CLN, MEN, and MLN. Conventional ENSO in the decaying phase is more likely to show wet and dry signals in comparison to the corresponding ENSO Modoki regimes.  Different performance of conventional ENSO and ENSO Modoki is a reflection of combined influence of the India monsoon and the WNP anti-cyclone. This study improved our understanding on the spatial variability of ENSO-induced precipitation during rainy season in China and the underlying causes. These results suggest that improved predictability can be achieved for rainy-season precipitation related to ENSO regimes. We suggest that further work should focus on the influence of interactive ENSO and other drivers on precipitation to evaluate and improve the predictive ability.

**6.   Data availability**

The daily precipitation, NOAA extended reconstructed SST, ENSO Modoki index (EMI) and the NCEP-NCAR reanalysis datasets used in this study are available for download under the following URLs:

– Daily precipitation: http://data.cma.cn/data/detail/dataCode/SURF_CLI_CHN_MUL_DAY_V3.0.html

– NOAA extended reconstructed SST: https://www.ncdc.noaa.gov/data-access/marineocean-data/extended-reconstructed-sea-
surface-temperature-ersst-v4

– EMI: http://www.jamstec.go.jp/frsgc/research/d1/iod/DATA/emi.monthly.txt

– NCEP/NCAR reanalysis data: https://www.esrl.noaa.gov/psd/data/gridded/data.ncep.reanalysis.html

*Author contributions.* Qing Cao, Zhenchun Hao and Feifei Yuan conceived the study. All authors contributed to writing the paper.

*Competing interests.* The authors declare no conflict of interest.

*Acknowledgements***:** Funding from the National Key Research Projects (Grant No. 2016YFC0402704) and the National Natural

Science Foundation of China (Grant No. 41371047) are gratefully acknowledged. Support from the China Postdoctoral Science

Foundation (Grant No. 2016M601711) and the Jiangsu Planned Projects for Postdoctoral Research Funds (Grant No. 1601027B)

are appreciated.